# Influence of Gelatin-Based Coatings Crosslinked with Phenolic Acids on PLA Film Barrier Properties

**Frédéric Debeaufort** [1,2,*] **, Julien Riondet** [1]**, Claire-Hélène Brachais** [3,4] **and Nasreddine Benbettaieb** [1,2]

1   IUT-Dijon-Auxerre, Department BioEngineering, University of Burgundy, 7 blvd Docteur Petitjean, 20178 Dijon, France; julien_riondet@outlook.fr (J.R.); nasreddine.benbettaieb@u-bourgogne.fr (N.B.)
2   UMR PAM A 02.102, Equipe PCAV, Univ. Bourgogne Franche-Comté, 1 Esplanade Erasme, F-21000 Dijon, France
3   ESIREM, University of Burgundy, Allée Savary, 21000 Dijon, France; claire-helene.brachais@u-bourgogne.fr
4   Laboratoire Interdisciplinaire Carnot de Bourgogne (ICB), UMR CNRS 6303, Univ. Bourgogne, Franche-Comté, 21078 Dijon, France
*   Correspondence: frederic.debeaufort@u-bourgogne.fr

**Abstract:** Single-use plastics are a major source of pollution and biodegradable polymers could be the best substitute, as they possess similar barrier and functional properties. Aiming at improving barrier properties and providing antioxidant bioactivity, PLA (PolyLactic Acid) films were coated with a crosslinked suspension of plasticized gelatin incorporating phenolic compounds. The coating process induced weak modifications of PLA properties due to plasticization by moisture and glycerol from the coating suspension. Indeed, a double glass transition was displayed. The water vapor barrier properties of the PLA-coated films were not significantly affected. Phenolic compounds induced a crosslinking of the gelatin network, slightly decreasing the moisture sensitivity and surface hydrophilicity. Therefore, the mechanical properties of PLA were maintained after coating and their barrier properties were highly improved, with up to a 600-fold reduction of the oxygen transfer rate. These results make possible new applications for oxidation-sensitive foods, and even for semi-moist foods.

**Keywords:** food packaging; polylactide; fish gelatin; tannic and gallic acids; oxygen and water transfers; surface hydrophilicity; thermal properties; moisture

## 1. Introduction

The main plastics responsible for visible pollution are single-use plastics (SUPs), of which 150 million tons circulate in marine environments for example, without forgetting all of the consequences of their degradation and their toxic impacts on fauna and flora. These plastics represented 50% of the waste collected on European beaches in 2016 and 29% of municipal waste in 2018 in France [1,2]. It is therefore necessary to find alternatives to these SUPs, as only 22% are recycled and 42% are collected and incinerated. This leaves about 36% of plastics that are dispersed in nature as macroplastic or micro–nano plastic particles [3–6].

This is a considerable challenge for the agri-food industry, which is the main user of UPCs, particularly for fresh and ultra-fresh products, and for catering [4]. There are currently two ways to respond to the problem of packaging pollution: (1) Create new recycling channels working on molecules such as polypropylene (PP), polystyrene and polyethylene; (2) design 100% biodegradable films [7–9].

Unfortunately, this last solution is not so easy to implement, because food packaging must meet many criteria and technical and sanitary requirements. Indeed, the packaging must have a protective function (mechanical resistance) against crushing and shocks. Then, we have the criterion of tightness, which is very important to avoid the dehydration or wetting of the product or its oxidation, for example. This criterion is based on the

phenomena of permeability and thus of barrier properties. External factors such as heat, moisture, oxygen and light can have an effect on the degradation of food because these factors will promote the development of micro-organisms such as bacteria and molds. Sometimes, a high permeability is required, for instance for fresh fruits and vegetables, for which the films may allow the penetration of oxygen and evacuate $CO_2$ for respiration. The permeability to gas, vapor, or light should then be tuned [10].

Poly lactic acid (PLA) films are 100% biodegradable, compostable and suitable for food contact. They are in the form of a resin of vegetable origin and are used in many fields such as textiles (fashion, upholstery, clothing), agriculture (nets, mats), hygiene (diapers, cloths), medicine (synthetic bones, disposable clothing) and food distribution (dishes, packaging). Poly lactic acid is obtained by fermentation of organic waste from the agricultural and food industries (corn for example) to extract the sugar. The sugar is then transformed into lactic acid, which, after purification, is polymerized into poly lactic acid. Today, PLA-based films are already marketed for dry food products or for fast-food type products (consumption on the spot). The problem is that this type of film cannot yet be used for products that are sensitive to external factors such as oxygen and especially humidity. These PLA-based packages are still very limited in use, mainly because of insufficient gas barrier properties [11].

The coating technique, which consists of depositing a thin layer of product on the film, is a simple and proven method to optimize the permeability of packaging materials including PLA [12]. Indeed, the strategy of coating PLA film by biopolymers such as chitosan, gelatin and other hydrocolloids to maintain biodegradability as well as improving barrier properties has been tested [13,14]. Gelatin is the second-most common source of proteins present in the animal world, having very interesting film-forming properties (pharmaceutical capsules) and gas barriers; it can also be used as a support for encapsulation of active molecules. However, what makes gelatin of interest for non-food applications is that it constitutes the major part of the waste of the marine industry, representing from 25% to 60% of the waste from aquatic products caught and processed, and can be used and modified to make biodegradable packaging materials and is compatible for food [2]. Different techniques of deposition of solutions or suspensions on films are available: Electrospray (spraying of liquid droplets from 100 nm to 10 µm), flexography (consisting of printing with flexible inkers) or simply by casting. All these techniques are followed either by drying or a stage of polymerization of the deposit by temperature, UV, etc. [15]. Previous works demonstrated that flexography coatings allows to provide bioactive properties to polyolefin-based films coated with chitosan and improving the barrier properties [16]. The preparation, application and drying at the lab scale allowed to reproduce the flexography industrial processing and gave satisfactory results related both to barrier and active properties without significantly affecting the polyolefin-support films.

This deposition of a gelatin-based suspension will therefore address the problems of transfer of materials but can also play on the mechanical strength of films. This coating also allows the incorporation of natural food additives such as antifungal and antimicrobial additives as well as antioxidants so as to provide bioactive functions to the films, as shown in a previous work [17]. Indeed, PLA films have been activated by the deposition of a film-forming solution containing antimicrobial additives or antioxidants, either by casting or by coating techniques [18]. Zein or other protein-based coatings did not allow to significantly improve the water vapor permeability of PLA films due to their hydrophilicity [19]. Chitosan coatings allowed to reduce the oxygen permeability of PLA films 2- to 5-fold [14,20] and 50-fold when coupled with $SiO_X$ [21]. Svagan et al. [22] succeeded to reduce the oxygen transfer of PLA film by 99% when chitosan and nanoclays were applied using a layer-by-layer technique, but the oxygen permeability reduction was moisture-dependent. Very few works dealt with gelatin-based coatings on PLA or the use of waste or by-products from the agro-food industry as the matrix of coating solutions and suspensions to be applied on industrial biodegradable films.

The objective of this study was to coat PLA films with different gelatin-based formulations and to study their mechanical, surface and barrier properties in order to optimize the properties and potential uses of PLA. Incorporation of tannic acid and gallic acid aimed at crosslinking the gelatin and providing antioxidant properties, respectively. Moisture's effects on the barrier properties were particularly studied.

## 2. Materials and Methods

### 2.1. Materials

A commercial fish gelatin (ref. CO-SP-004, Louis François, Marne-La-Vallée, France) with a Bloom degree = 200, viscosity = 3.5–4.5 mPa·s at 60 °C, a concentration of 6.67% in water at pH = 5.8 and a pI = 4.8–5.3 was used as the biopolymeric matrix of the coatings. Anhydrous glycerol (gly) of 98% purity (Fluka-Fisher Scientific SAS, Illkirch, France) was used as plasticizer.

A commercial and biodegradable semi-crystalline poly-lactic acid (PLA) film (Nativia-NTSS, Taghleef Industries, San Giorgio di Nogaro, Italy) was used as a reference film for optimization of its functional properties. The Nativia-NTSS PLA films were treated by corona discharge on both sides during industrial manufacturing. This treatment facilitates the adhesion of the hydrophilic active coating.

The samples were cut to the exact dimensions (50 cm × 20 cm) for further coating processes.

Gallic acid (minimum purity 98%, Mw = 170.12 g·mol$^{-1}$, MV = 97.3 cm$^3$·mol$^{-1}$, density = 1.749 g·cm$^{-3}$, MP = 251 °C, BP = 501 °C, LogP = 0.964, pKa = 4.09 in water or aqueous solution, solubility in water = 11.5 g·L$^{-1}$ at 20 °C, data from www.ChemSpider.com (accessed on 21 April 2021)) and tannic acid (minimum purity 98%, Mw = 1701.2 g·mol$^{-1}$, MV = 799.1 cm$^3$·mol$^{-1}$, density = 2.1 g·cm$^{-3}$, MP = 218 °C, LogP = 9.537, pKa in water or aqueous solution = 10, solubility in water = 2.85 g·L$^{-1}$ at 20 °C, data from www.ChemSpider.com (accessed on 21 April 2021)) were from Sigma-Aldrich Chimie SARL (Saint-Quentin Fallavier, France). The two phenolic acids were selected because they have natural antioxidant activities but also act as protein cross-linkers. Tannic acid is considered a food additive (E181) without any restriction of use or maximum daily intake according to the EFSA authority [23]. In the case of gallic acid, several studies have shown that at concentrations below 1000 mg/kg of food it does not present a risk to the consumer [24]. These two phenolic compounds have been selected as they are in high quantity in waste from the paper/wood industries; however, the amount used in the film-forming suspension is ten times lower for tannic acid because of its lower solubility and higher cross-linking capacity.

Magnesium nitrate (Mg(NO$_3$)$_2$), sodium bromide (NaBr), sodium chloride (NaCl) and potassium chloride (KCl) (Sigma-Aldrich Chimie SARL, Saint-Quentin Fallavier, France) were used to prepare saturated salt solutions to fix the relative humidity for water vapor sorption measurements, respectively at 33%, 58%, 75% and 84% at 25 °C. Phosphorus pentaoxide (P$_2$O$_5$, Sigma-Aldrich Chimie SARL, Saint-Quentin Fallavier, France) was used to fix the relative humidity at ≈0% to dry the sample before sorption or thermal analysis.

### 2.2. Coatings and Coated Films Preparation and Thickness Characterization

Several formulations of coatings were previously developed to optimize their adhesion to PLA films and their antioxidant properties [17].

A stock solution of 15% (*w/w*) gelatin in distilled water at 50 °C was prepared. The temperature of 50 °C was maintained throughout the manufacturing process of the different samples until the casting or coating steps. After 30 min of moderate magnetic stirring (100 rpm), glycerol (10% of the gelatin weight) was added and the mixture was stirred for 10 min. The gelatin-gly film-forming master suspension prepared in this way was then either poured directly into square-section Petri dishes (13 × 13 cm$^2$) in order to study the standalone coatings, or spread on the PLA films (process described later).

To prepare the coatings and coated films with 5% gallic acid (named gelatin-gly + 5% gallic ac.), the master film-forming suspension was added to the plasticized stock solution

(5% of the gelatin weight) followed by a 20 min stirring at 50 °C. The concentration of the gallic acid was optimized according to its solubility in aqueous solution.

According to the same protocol, tannic acid was added at a concentration of 0.5% to the plasticized stock suspension and kept for 30 min under stirring to obtain the gelatin-gly-tanned coatings. Tannic acid is considered more for its action as a tanning agent than for its antioxidant properties. Indeed, both its high tanning capacity and very low solubility in aqueous liquids limited the concentration used.

In order to highlight possible synergistic or antagonistic effects between the two phenolic acids, a formulation including gallic acid was prepared after tanning with tannic acid. Gallic acid was therefore added to the plasticized and tanned parent suspension up to 5% (gelatin mass) and kept for 20 min under stirring.

Once the gelatin suspensions were ready, we immediately proceeded to the coating of the PLA film. This was carried out by maintaining the spreading temperature at 50 °C (suspension and all the equipment used) to avoid the gelatin from gelling.

A thin layer chromatography applicator (Desaga Brinckmann, DS200/0.3 TLC, Heidelberg, Germany) was used to reproduce the wet coating technique on a laboratory scale. The PLA films were first fixed on a plexiglass plate placed on the applicator template. The gelatin solution was poured into the applicator (set to have a liquid thickness of 250 μm) and then spread at a constant speed of about 20 cm/s. The coated PLA films are then left to dry for about 24 h under ambient conditions (25 ± 2 °C and 30% ± 5% relative humidity). These conditions allow to reproduce a fast drying at 90 °C in less than 10 s (unpublished data).

The standalone coatings were obtained by pouring 10 g of solution into Petri dishes of square section ($13 \times 13$ cm$^2$) in order to obtain a thickness of about 60 μm. Once poured, they were left to dry under the same conditions as for the coated PLA films, and then the standalone coatings were peeled from the support.

All samples (coated films and coatings) were then stored at 50% relative humidity and 25 °C.

Films thickness was measured using a digital thickness gauge (PosiTector 6000, DeFelsko Corporation, Ogdensburg, NY, USA). Five measurements were taken from each film sample at different locations, one at the center and four from the perimeter.

### 2.3. Rheology of Coating-Based Suspensions

The rheological properties of the film-forming suspensions were characterized using a dynamic rheometer (Anton Paar, MCR 302, Anton-Paar, Graz, Austria). Measurements of the storage (elastic) modulus G′ and loss modulus G′ (viscous modulus) were performed on the film-forming suspensions previously heated to 50 °C.

The measurements were made with a shear modulus with a plane/plane disk geometry of diameter 25 mm (pp50), with a force applied to the sample of less than 0.03 N and a dynamic displacement (oscillation) of 1 mm, over a temperature range of 50 and 10 °C with a rate of 1 °C·min$^{-1}$ and a frequency of 1 rad·s$^{-1}$. Only the values at 40 °C (liquid state) and 10 °C (gelled state) were considered and discussed.

The storage modulus (G′) and loss modulus (G″) were determined. The storage (elastic) modulus is defined as the ratio of the elastic (reversible) stress to the strain, whereas the loss modulus is the ratio of the viscous component to the stress; G″ is related to the ability of the material to dissipate the stress by heat. The results obtained provide information on the structural properties of the sample, the cross-linking and any changes at the molecular level.

### 2.4. Moisture Sorption Isotherm and GAB Modeling

The water vapor sorption isotherms of the films and coatings were determined by the microclimate method [25]. Approximately 600 mg of the sample were pre-dried on silica gel for 20 days at 25 °C. They were then stored until equilibrium (constant mass) in ventilated chambers containing different relative humidity levels (33%, 58%, 75% and 84.4% RH) fixed by saturated salt solutions. After equilibrium, the water content of the

films was determined by oven drying at 102 °C for 12 h. The equilibrium water contents are expressed as grams of water per 100 g of dry matter (% b.d).

The Guggenheim–Anderson–de Boer (GAB) equation is the most widely used model in European food science research laboratories [25]. The equation is applicable for water activities between 0.05 and 0.95 and can be formulated according to the following Equation (1) [26].

$$W = \frac{M_0 \times C \times K \times A_w}{(1 - K \times A_w) \times (1 - K \times A_w + C \times K \times A_w)} \tag{1}$$

where $M_0$ is the water content of the monolayer (g water/100 dry basis), $A_w$ is the water activity, C is the Guggenheim constant in relation to the heat of sorption of the monolayer and K is the correction factor for multilayers.

The Guggenheim–Anderson–de Boer (GAB) model represents an extension of the Langmuir and BET theories over a wider range of relative humidity. The GAB theory assumes localized physical adsorption in multilayers with non-lateral interactions [27]. The first water layer $M_0$ is very strongly bound to the absorbing substrate. The prediction of the value of $M_0$ is therefore very important, as it generally corresponds to the threshold of spoilage of food products [26,28]. The constants C and K in the GAB equation are functions of the heat of sorption of the monolayer (C) and the correction factor for the heat of sorption of the subsequent layers (K) absorbed on the solid substrate according to the following equations [29].

$$C = C_0 \times e^{\left(\frac{H_w - H_{M0}}{R \times T}\right)} \tag{2}$$

$$K = K_0 \times e^{\left(\frac{H_w - H_q}{R \times T}\right)} \tag{3}$$

where $H_w$ is the heat of condensation of pure water, $H_{M0}$ is the total sorption heat of the first layer and $H_q$ is the total sorption heat of the multilayers, $C_0$ and $K_0$ are the re-exponential factor of the Arrhenius-type equation giving the proportionality of the K and C parameters to the temperature, R is the perfect gas constant, and T the temperature in Kelvin.

*2.5. Thermal Analysis*

Differential scanning calorimetry (DSC, TA instruments, Discovery Series, New Castle, UK) was used to study the thermal properties of the films and coatings, in particular to assess the impact of the coating on the properties of the PLA. These measurements reproduced the procedure detailed by Benbettaieb et al. [30], except that the heating and cooling cycles were adapted to the coated films:

-   Stabilization (equilibration) at 25 °C for 5 min and cooling from 25 to −80 °C at the rate of 10 °C/min$^{-1}$ followed by isothermal holding for 5 min;
-   First heating from −80 to +120 °C at a rate of 10 °C/min$^{-1}$, immediately followed by a second cooling to −80 °C at a rate of 10 °C/min$^{-1}$, followed by isothermal stabilization for 5 min;
-   Second heating from −80 to +180 °C at a rate of 10 °C/min$^{-1}$ and finally cooling (drop) to 25 °C.

The glass transition temperature, $T_g$ (°C), along with the related specific heat change ($\Delta C_p$, W·g$^{-1}$) of each sample, was determined from the second heating cycle using TA Universal Analysis 2000 software (version 4.5 A, TA instruments, New Castle, DE, USA). The melting temperature, $T_m$ (°C), of PLA as well as the change in melting enthalpies ($\Delta H$, J·g$^{-1}$) were also determined. All films were previously equilibrated to 0% RH and 25 °C before the DSC experiments.

*2.6. Mechanical Properties Determination*

The mechanical properties of the different specimens were characterized with a universal tension-compression machine (TA-HD Plus+, Stable Micro-Systems, Godalming SU, UK). The tests consisted of applying two opposing forces to a sample. Coated films or coat-

ing samples were cut into a rectangle ($2.54 \times 8$ cm$^2$) using a standardized double-bladed precision cutter to obtain tensile specimens with exact width and parallel sides and no defects along their length. The ends of the specimens were clamped between the jaws of the tension testing machine before being subjected to uniaxial stretching (300 N load cell) at an imposed and constant speed of 50 mm/min until failure. The measurements were performed at room temperature ($25 \pm 2$ °C) and at a RH of 50%, and nine repetitions of each formulation were tested. The measurement conditions were adapted for biomaterial-based multilayer films based on ISO 527-1 [31]. The forces were recorded by the load cell located on the fixed part of the machine. Each material was characterized by a stress-strain curve from which the force at break or tensile strength at break (TS), elongation at break (EAB) and Young's modulus (elastic component, YM) were determined. The tensile strength and elongation of the first fracture were also considered and correspond to the delamination (separation) of the coating from the PLA film support.

### 2.7. Surface Property Measurement

Goniometry allows to measure physical–chemical characteristics of an interface between two phases (liquid/solid) using different measurements: Contact angle, surface energy or SFE (drop method), surface tension (hanging drop method) and adhesion energy (drop method and hanging drop) [32]. Surface properties will also influence the transfer through the films by facilitating or not the spreading of liquids on the surface.

Measurement of the water contact angle was performed using a Kruss goniometer (DSA30, Kruss GmbH, Villebon, France). A drop of distilled water (~2 μL) was automatically deposited on the film or coating surface. Using a camera and the Kruss-Advance operating software (version 2) that analyzed the drop shape, the contact angle and drop volume were measured for 2 min to evaluate the wetting kinetics (drop spreading speed). Our tests were conducted in a controlled room at a temperature of $25 \pm 2$ °C and a relative humidity of $50\% \pm 1\%$, and at least 10 replicates were made.

### 2.8. Moisture and Oxygen Permeation Analysis

The water vapor transmission coefficient (WVTRP) was determined using the gravimetric cup method described in ISO 2528 [33] and adapted to hydrophilic biopolymers by Debeaufort et al. [34]. A relative humidity gradient of 30%–100% and a temperature of $25 \pm 1$ °C were used. Prior to measurements, all film samples were equilibrated at $25 \pm 1$ °C and 50% relative humidity for at least 72 h. The test film was placed on a glass cell containing pure water (RH = 100%) and stored in a ventilated chamber regulated at 25 °C and 30% (KBF 240 Binder, Tuttlingen, Germany). The WVTR (g·m$^{-2}$·s$^{-1}$) was calculated from the linear part of the mass change (weight loss) versus time curve according to the following equation (Equation (4)):

$$\text{WVTR} = \frac{\Delta m}{A \times \Delta t} \tag{4}$$

where $\Delta m / \Delta t$ is the mass loss per unit time (g·s$^{-1}$) and A is the area of the film exposed to moisture transfer ($9.08 \times 10^{-4}$ m$^2$). Five replicates for each sample were performed.

Regarding the oxygen transfer through the films, a Brugger GTT permeameter (Brugger Feinmechanik GmbH, Munich, Germany) was used to measure the gas transmission through the thin film materials by the manometric (pressure difference) method, according to ISO 15105-1 [35]. The film or coating samples were placed in a cell where one compartment was flushed with oxygen and where the relative humidity was regulated, and the other compartment was subjected to a high vacuum at first, and then the pressure was monitored over time. The temperature of the measuring chamber was regulated at 25 °C. The software gives a graphical representation of the oxygen transmission rate (OTR cm$^3$·m$^{-2}$·s$^{-1}$) as a function of time until the steady state of permeation is reached where the OTR value is recorded. The measurements on each of the samples (films and coat-

ings) were performed at three different humidity levels (10%, 50% and 85%) and repeated at least three times.

### 2.9. Statistical Analysis

Statistical analysis of data was performed with SPSS 13.0 software (Stat-Packets Statistical analysis Software, SPSS Inc., Chicago, IL, USA). Using this software, analysis of variance (ANOVA) was carried out in order to determine the significant difference through the multiple comparisons of means. The least significant difference (LSD) mean comparison test was used at the significance level of 95% ($p$-value $< 0.05$) for all characterizations. In the case of thermal analysis (DSC), three repetitions were performed on the control film (PLA NTSS) in order to calculate the average relative error, as follows: average relative error = (standard deviation/mean) $\times$ 100. The average relative error (%) was applied to the values of other samples when only a single measurement could be done.

## 3. Results and Discussions

### 3.1. Rheology of the Gelatin-Based Film-Forming Suspensions

Figure 1 shows the evolution of the appearance of the suspension at the different stages of preparation. After the successive dispersion and solubilization of the gelatin and the glycerol, the suspensions were clear and slightly viscous. However, when gallic acid was added at 5% into the gelatin-glycerol suspensions, these became cloudier and apparently more viscous. The addition of the tannic acid at 0.5% induced a similar behavior. This anisotropic system confirms that the phenolic acids interact with the gelatin to form a more organized network. Indeed, Le et al. [36] demonstrated that interactions occurring between fish gelatin and phenolic compounds modified some properties but are of week energy (hydrogen bonds) and could be disrupted by Sodium Docecyl Sulfate (SDS) prior to SDS-PAGE electrophoresis. This also means that there was no covalent bond generated and thus the phenolic acids did not permanently crosslink (covalent bonds) the gelatin chains.

| 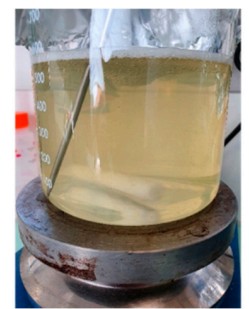 | 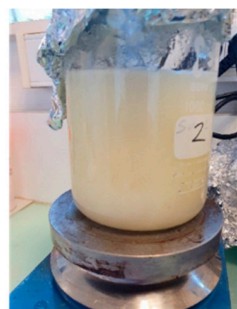 | 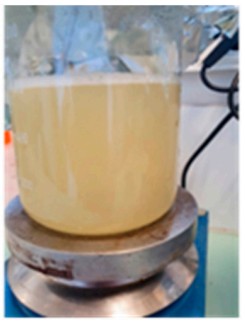 | 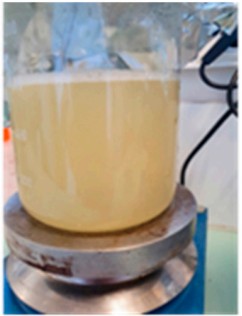 |
|---|---|---|---|
| Master batch of gelatine-glycerol suspension after 30 min stirring at 50 °C | After the addition of only the gallic acid and 30 min stirring | After the addition of only the tanning acid and 30 min stirring | After addition of tannic acid and stirring for 30 min followed by gallic acid and 30 min stirring |

**Figure 1.** Images of the gelatin suspension at each step of its preparation, after solubilization of tannic and gallic acids.

The dynamic rheology curves (Supplementary Data) allowed to determine graphically the temperatures of the melting and gelation points of the gelatin-based suspensions before they were poured into Petri dishes or cast onto PLA films (coating process) (Figure 2). Furthermore, the values of the storage modulus G′ and loss modulus G″ at 10 and 40 °C were determined in order to compare the two physical states (gelled and liquid) of the suspensions and the effects of the two phenolic acids' incorporation on their rheology.

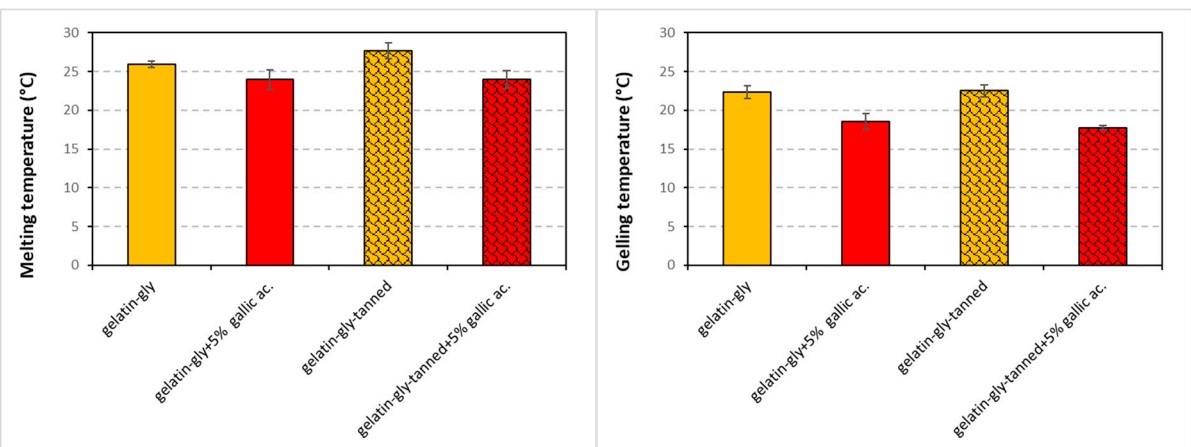

**Figure 2.** Influence of the composition of gelatin suspensions on their melting and gelling temperatures.

The melting temperature of the fish gelatin-glycerol suspension was 25.9 °C and was similar of the values measured by many authors for fish gelatins, which range from 23 to 29 °C according the species [37].

During the heating phase, the gallic acid tends to lower the melting temperature of the gel. Indeed, we found a melting temperature of 25.9 °C for the gelatin-gly solution in contrast to 23.9 °C for the solution containing gallic acid. Conversely, when we used tannic acid alone, the temperature was increased (27.6 °C). The two phenolic acids thus have an opposite effect, which is confirmed when we observe the histogram of suspension coat 4 (gelatin-gly-tanned + 5% gallic ac.). It appears that the addition of 0.5% tannic acid or 5% gallic acid induces modifications/interactions at the molecular level of the gelatin network. This has already been displayed by Benbettaieb et al. [26] for a fish gelatin-glycerol suspension containing some phenolic acids.

Regarding the cooling phase (gelation temperature), the same behavior as for the melting temperature was observed but more significantly. The shift observed between the melting and gelation temperature is due to the measurement technique, and in particular, due to the rate of temperature variation that is attributed to the hysteresis.

In the gel state (at 10 °C), gallic acid caused a loss of elasticity and stiffness (viscosity) while the tannic acid seems to have had no significant effect (Figure 3) due to its low concentration. In the liquid state (40 °C), only the synergic effect of both tannic and gallic acids had an effect on the elastic component (elasticity mode). When the suspension contained both acids, a decrease in viscosity and elasticity was observed. This means that the interactions between the gelatin chains were disturbed when these acids were added. The power of tannic acid, independently of gallic acid, can be explained by the fact that it provides a "cross-linking" function that allows to bind proteins more efficiently, which is well reflected by an increase in viscosity (loss modulus) and a loss of elasticity (storage modulus). Indeed, tannic acid is a "polymer" of GA (decagalloyl glucose), composed of a benzenic ring "polymerized" (ester links) with 10 gallic acid units. Therefore, TA has 10 carboxyl groups and 25 hydroxyl groups that are able to interact with functional groups of gelatins (in adequate pH, $O_2$ concentration and temperature), compared to gallic acid that has only one COOH and three OH groups. According to Zhang et al. [38], lysine, arginine and histidine could possibly react with the phenolic reactive sites of gelatin as well as the carboxyl and hydroxyl groups, even if pH conditions are not optimized, to form hydrogen bonds that may affect physical properties.

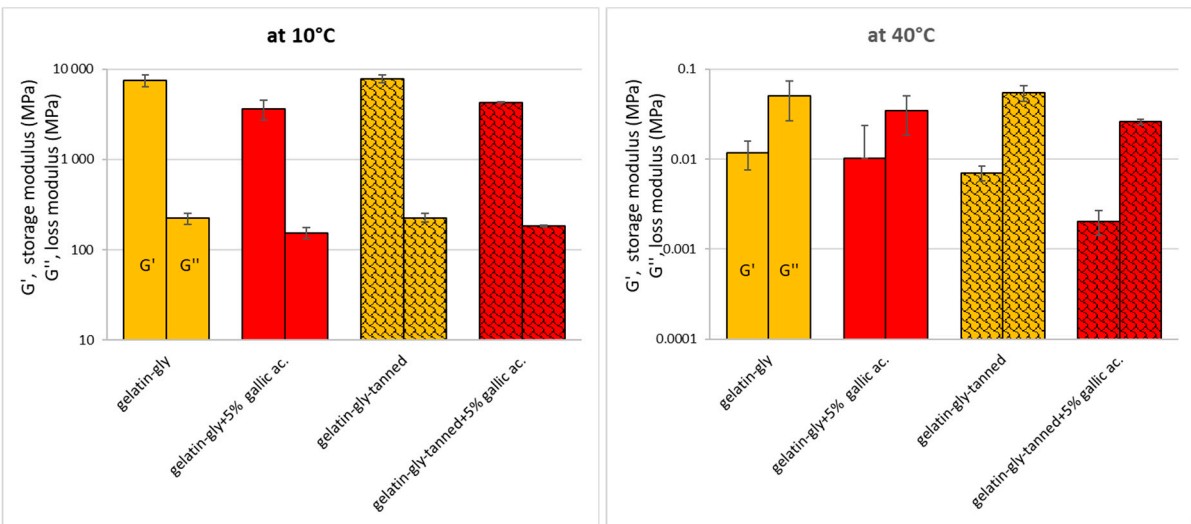

**Figure 3.** Influence of the composition of gelatin suspensions on the elastic and loss modulus at 10 °C (solid state) and 40 °C (liquid state).

The rheology properties influence the mechanical properties of coated films because the structure of the gel before drying conditions the organization of the solidified and dried network.

### 3.2. Moisture Sorption Isotherms of Films

The water vapor sorption isotherms at 25 °C of the standalone coatings are given in Figure 4 and they display the characteristic form of type IV isotherms according to the International Union of Pure and Applied Chemistry (IUPAC) nomenclature [39]. They are very similar to those determined by several authors for gelatin films, and the differences in water content are explained by the origin of the gelatin (bovine or fish) and by the plasticizer content [40,41]. The addition of phenolic acids slightly reduces the water sorption, which can be attributed to the interactions between acids and gelatin and thus to a more structured organization of the gelatin network. The sites of water vapor fixation by hydrogen bonds are usually mobilized by phenolic acids.

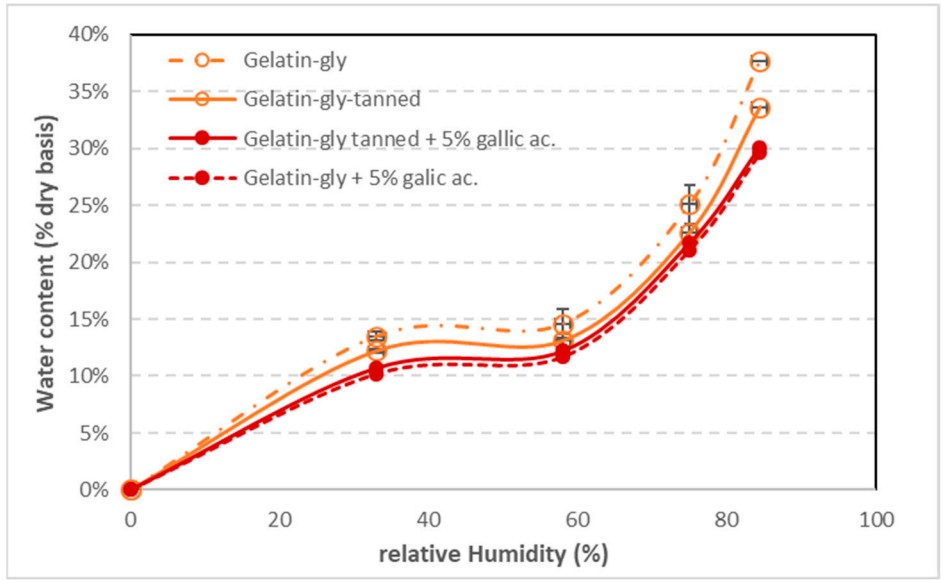

**Figure 4.** Moisture sorption isotherms of standalone coatings measured at 25 °C.

Our PLA film isotherms (Figure 5) are similar to those determined by Henton et al. [12] and Mucha and Ludwiczak [42] and show a type II or IV isotherm depending on the authors. De Oliveira Pizzoli et al. [43] determined the isotherms of extruded PLA films containing thermoplastic starch and various levels of gelatin. As we also observed, the amount of water absorbed by the coated films is proportional to the amounts of the constituents (Figure 5). Indeed, the isotherms of coated PLA films, whatever the coating composition, have water contents that are directly proportional to the relative thicknesses (and thus weight) of the PLA film and the coating layers. Coating composition seems not to significantly affect the moisture sorption isotherms of coated PLA.

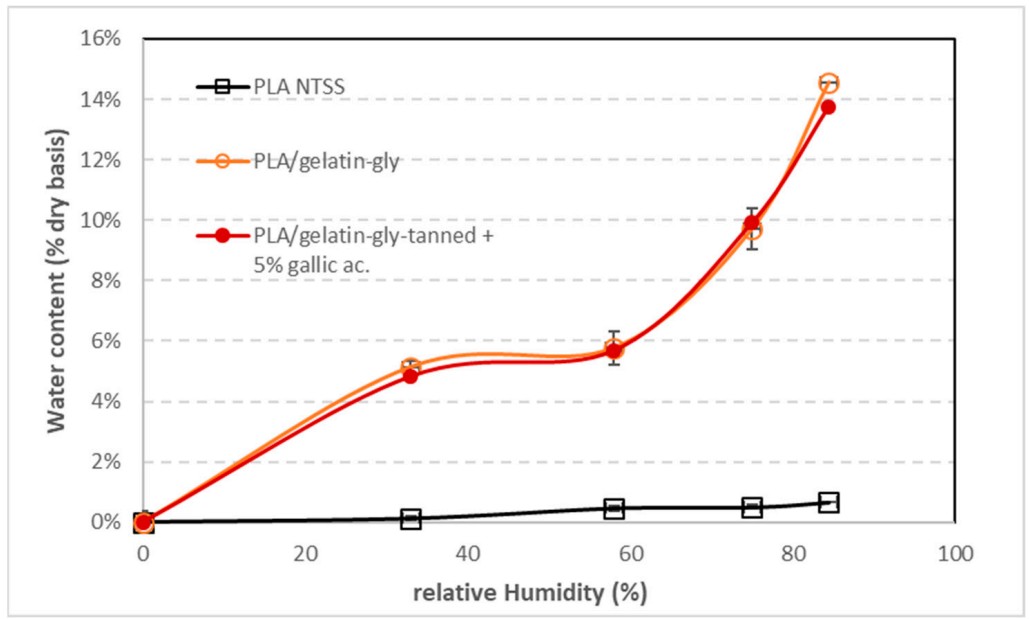

**Figure 5.** Moisture sorption isotherms of PLA films and of coated PLA films (PLA/gelatin-gly and PLA/gelatin-gly tanned + 5% gallic ac.), measured at 25 °C.

The GAB model was applied to all of the isotherms and the value of the monolayer ($M_0$) and the constants of the model are given in the Table 1. The water content of the monolayer absorbed by PLA was 10 times lower than that of the coatings. As the structures are dense, the specific surface area of adsorption is reduced and thus the $M_0$. Phenolic acids have a cross-linking role; they slightly reduce the water affinity of the films and thus the water content of the monomolecular layer. The GAB model fits well to experimental values with $R^2 > 0.97$ and very low RMSE. The constants C and K in the GAB equation are functions of the heat of sorption of the monolayer (C) and the correction factor for the heat of sorption of subsequent layers (K) absorbed on the solid substrate. C and K values are only significant between neat PLA films and coatings, but there is no significant difference between the different coatings and coated films. It can be supposed that the heat of sorption is not affected by the change in structure due to phenolic compounds.

The isotherms of films and coatings confirmed the moisture dependence of the gelatin layer on moisture content according their composition, which may affect the structural (thermal and mechanical) and barrier properties of the coated PLA films.

**Table 1.** GAB model values of samples: $M_0$ monolayer water content, C and K constants, and the RMSE of the GAB model fitting (at 25 °C).

| Films and Standalone Coatings | $M_0$ (g/g·dm) | C | K | RMSE |
|---|---|---|---|---|
| PLA NTSS | 0.013 [a] | 0.3042 | 0.9061 | $2.42 \times 10^{-7}$ |
| **Coatings** | | | | |
| Gelatin-gly | 0.120 [c] | 0.6667 | 2.1338 | $5.10 \times 10^{-3}$ |
| Gelatin-gly-tanned | 0.117 [c] | 0.6569 | 2.0322 | $4.10 \times 10^{-3}$ |
| Gelatin-gly 5% gallic ac. | 0.113 [c] | 0.6464 | 1.9514 | $2.29 \times 10^{-3}$ |
| Gelatin-gly-tanned + 5% gallic ac. | 0.117 [c] | 0.6654 | 1.9643 | $2.50 \times 10^{-3}$ |
| **Coated Pla** | | | | |
| PLA/gelatin-gly | 0.089 [b] | 0.5196 | 1.4600 | $0.70 \times 10^{-3}$ |
| PLA/gelatin-gly-tanned | 0.103 [b,c] | 0.5769 | 1.6345 | $1.60 \times 10^{-3}$ |
| PLA/gelatin-gly 5% gallic ac. | 0.084 [b] | 0.4399 | 1.4316 | $2.31 \times 10^{-3}$ |
| PLA/gelatin-gly-tanned + 5% gallic ac. | 0.088 [b] | 0.5530 | 1.4535 | $0.47 \times 10^{-3}$ |

[a], [b], [c]: values with the same letter in each column are not significantly different at *p*-level = 0.05.

*3.3. Thermal Properties*

Thermal analysis was very repeatable, and thus measurements were only triplicated on the dry PLA NTSS control films (Table 2). The variance (relative error) determined on the $T_g$ and melting temperatures, $\Delta C_p$ of $T_g$, and enthalpy of melting of the PLA NTSS was applied to all other samples (standalone coatings and coated PLA films).

As already demonstrated by several authors, the Tg of PLA decreases with the water content due to the plasticization, varying from 66.3 °C at 0% RH down to 61.4 °C at 85% RH. Another $T_g$ appeared when the PLA films were exposed to moisture, decreasing from 58.7 to 56.1 °C and increasing the RH. This let us suppose that the composition or structure of the PLA film is inhomogeneous, probably due to the Corona treatment of the film surface during film processing. This was also observed by Rocca-Smith et al. [11] on similar commercial PLA NTSS films. The impact of humidifying PLA films had the same influence as the coating process with gelatin-based coatings. Indeed, wet coating induced an increase of the moisture content by about 0.05% average (data not given), on a thickness portion of the PLA film at the coated surface. Indeed, all coated films presented a double $T_g$, one around 65–67 °C corresponding to unmodified/unwetted PLA and the one of the plasticized PLA zone ranging approximately from 55 to 61 °C.

The melting temperature and melting enthalpy of PLA were not significantly modified by the moisture level nor by the coating.

Dealing with the coatings, the gelatin-glycerol standalone coating exhibited two $T_g$ values; one varied from −7.9 to −18.3 7 °C and the other ranged from 24.2 to 29.9 °C, depending on the composition. The two $T_g$ values are related to an excess of plasticizer (glycerol). Indeed, as displayed by Chaudhary et al. [44] for starch-based films, plasticized biopolymers such as polysaccharides or proteins sometimes are organized with a glycerol-enriched phase with a low $T_g$ value and have high molecular mobility and a less enriched phase with a higher $T_g$ value, as we observed. The gelatin structure is very sensitive to plasticizer content and could exhibit a type of multiphasic systems that could also explain the two $T_g$ values observed [45]. The addition of phenolic acids favored reticulation and consequently some sites available for the gelatin–glycerol interaction were not available. Therefore, the glycerol-enriched phases tended to absorb more water and drop the $T_g$ value. Indeed, when gallic acid and/or tannic acid or both acids were added, the $T_g$ decreased from −7.9 to −12.8, −12.2 and −18.3 °C, respectively. The higher the reticulation was (by the couple of both tannic and gallic acid), the more water was excluded from gelatin networks and the lower its plasticization was efficient ($T_g$ increased). Additionally, the addition of phenolic acids induced a decrease of the melting temperature of gelatin because of a less homogeneous system (the melting peak is much broader) due to the crystallization disturbance and polydispersity of the crystallin zone size, but the enthalpy of melting was not significantly affected. Similar trends have been observed by Sobral et al. [46].

**Table 2.** Thermal characteristics of PLA films according the relative humidity, and of both the standalone coatings and PLA coated films (L, thickness; RH, relative humidity; $T_g$, glass transition temperature; $\Delta C_p$, the specific calorific capacity variation of $T_g$; $T_{melt}$, melting temperature and $\Delta H_{melt}$, the melting enthalpy).

| PLA Films, PLA Coated Films and Standalone Coatings | L (μm) | RH (%) | Phenomena Attributed to PLA | | | | | | Phenomena Attributed to Gelatin-Based | | | | | |
|---|---|---|---|---|---|---|---|---|---|---|---|---|---|---|
| | | | $Tg_1$ (°C) | $\Delta Cp_1$ (W/g) | $Tg_2$ (°C) | $\Delta Cp_2$ (W/g) | $T_{melt}$ (°C) | $\Delta H_{melt}$ (J/g) | $Tg_1$ (°C) | $\Delta Cp_1$ (W/g) | $Tg_2$ (°C) | $\Delta Cp_2$ (W/g) | $T_{melt}$ (°C) | $\Delta H_{melt}$ (J/g) |
| *PLA NTSS replicates* | *25 ± 1* | *0* | */* | */* | *64.8* *67.6* *66.6* | *0.0301* *0.0289* *0.0389* | *149.7* *151.2* *150.6* | *35.9* *26.1* *33.4* | - | - | - | - | - | - |
| PLA NTSS average | 25 ± 1 | 0 | / | / | 66.3 ± 1.4 [a] | 0.0326 ± 0.0055 [a] | 150.0 ± 0.8 [a] | 31.8 ± 5.1 [c] | - | - | - | - | - | - |
| PLA NTSS | 25 ± 1 | 58 | 56.7 ± 1.2 [a] | 0.0141 ± 0.0024 [b] | 63.7 ± 1.4 [a] | 0.0317 ± 0.0053 [a] | 151.2 ± 0.8 [a] | 19.7 ± 3.2 [a] | - | - | - | - | - | - |
| PLA NTSS | 25 ± 1 | 75 | 57.7 ± 1.2 [a] | 0.0159 ± 0.0027 [b] | 63.9 ± 1.4 [a] | 0.0282 ± 0.0047 [a] | 150.4 ± 0.7 [a] | 24.4 ± 3.9 [b] | - | - | - | - | - | - |
| PLA NTSS | 25 ± 1 | 84 | 56.1 ± 1.2 [a] | 0.0187 ± 0.0031 [b] | 62.4 ± 1.3 [a] | 0.0345 ± 0.0058 [a] | 150.5 ± 0.8 [a] | 23.6 ± 3.8 [b] | - | - | - | - | - | - |
| **Coated PLA** — PLA/Gelatin-gly | 40.8 ± 3.6 [a] | 50 | 55.7 ± 1.2 [a] | 0.0102 ± 0.0017 [a] | 67.2 ± 1.4 [a] | 0.0322 ± 0.0054 [a] | 149.2 ± 0.7 [a] | 20.8 ± 3.3 [a] | −7.7 ± 0.5 [c] | 0.0300 ± 0.0050 [b] | 40.2 ± 1.9 [b] | 0.0230 ± 0.0038 [a] | ND | ND |
| PLA/Gelatin-gly-tanned | 79.5 ± 13.5 [b] | 50 | 55.6 ± 1.2 [a] | 0.0261 ± 0.0044 [c] | 64.9 ± 1.4 [a] | 0.0460 ± 0.0077 [b] | 149.1 ± 0.7 [a] | 14.7 ± 2.4 [a] | −7.2 ± 0.7 [c] | 0.0487 ± 0.0081 [c] | 34.8 ± 1.2 [b] | 0.0431 ± 0.0072 [c] | ND | ND |
| PLA/Gelatin-gly 5% gallic ac. | 64.6 ± 15.1 [a] | 50 | 56.4 ± 1.2 [a] | 0.0106 ± 0.0018 [a] | 67.3 ± 1.4 [a] | 0.0340 ± 0.0057 [a] | 148.9 ± 0.7 [a] | 29.9 ± 4.8 [b] | −8.6 ± 0.5 [c] | 0.0361 ± 0.0060 [b] | 41.6 ± 0.9 [b] | 0.0273 ± 0.0046 [b] | ND | ND |
| PLA/Gelatin-gly-tanned + 5% gallic ac. | 44.6 ± 10.2 [a] | 50 | 60.8 ± 1.3 [b] | 0.0357 ± 0.0060 [d] | 66.2 ± 1.4 [a] | 0.0281 ± 0.0047 [a] | 149.4 ± 0.7 [a] | 22.5 ± 3.6 [b] | −6.7 ± 0.5 [c] | 0.0288 ± 0.0048 [b] | 38.9 ± 1.8 [b] | 0.0367 ± 0.0061 [b] | ND | ND |
| **Coatings** — gelatin-gly | 83.4 ± 8.3 [b] | 50 | - | - | - | - | - | - | −7.9 ± 0.2 [c] | 0.0264 ± 0.0044 [b] | 24.2 ± 1.5 [a] | 0.0181 ± 0.0030 [a] | 156.9 ± 0.8 [b] | 38.0 ± 6.1 [a] |
| gelatin-gly-tanned | 59.8 ± 7.2 [a] | 50 | - | - | - | - | - | - | −12.8 ± 0.7 [b] | 0.0253 ± 0.0042 [b] | 27.8 ± 0.9 [a] | 0.0188 ± 0.0031 [a] | 139.0 ± 0.7 [a] | 34.2 ± 5.5 [a] |
| gelatin-gly 5% gallic ac. | 85.6 ± 5.9 [b] | 50 | - | - | - | - | - | - | −12.2 ± 0.8 [b] | 0.0175 ± 0.0029 [a] | 26.0 ± 1.6 [a] | 0.0206 ± 0.0034 [a] | 140.1 ± 0.7 [a] | 29.2 ± 4.7 [a] |
| gelatin-gly-tanned + 5% gallic ac. | 90.4 ± 8.7 [b] | 50 | - | - | - | - | - | - | −18.3 ± 1.4 [a] | 0.0148 ± 0.0025 [a] | 29.9 ± 1.6 [a] | 0.0340 ± 0.0057 [b] | 143.2 ± 1.7 [a] | 36.5 ± 5.8 [a] |

[a], [b], [c]: values with the same letter in each column are not significantly different at *p*-level = 0.05.

The coated film displayed the two $T_g$ values of both the PLA and gelatin. The PLA $T_g$ values and melting temperatures and enthalpy were not modified compared to hydrated PLA NTSS control films. This confirms that the PLA was hydrated during the coating process. Dealing with the coating layer, as the PLA absorbed a part of the moisture of the gelatin-based layer, and the $T_g$ values of the gelatins were higher than that of the standalone coatings. This is in accordance with what was observed from the sorption isotherm that displayed a water content below the proportionality regarding the layer weights or thicknesses, particularly when phenolic acids were added. The melting of the gelatin could not be observed, as it was overlapped by that of the PLA.

The thermal analyses confirmed that the gelatin-based layers were structurally modified by the phenolic acids and may affect the mechanical and barrier properties.

### 3.4. Mechanical Properties of Coated and Uncoated PLA Films

As a matter of principle, the more resistant (stiffness, elongation, etc.) the material is, the higher the force to be applied should be during measurement, and vice versa. The effect of phenolic compounds on the deformation (% of elongation at first fracture) and the maximum strength at first fracture of the coated films were measured. The raw curves (data not given) show that the coating broke very early, prior to the PLA film breaking, from 3% to 4% of the deformation, which corresponds to the first fracture (Figure 6). Indeed, the gelatin layers deposited on PLA were very thin and broke very quickly, and they do not deform in the same range (30 times less than PLA, Figure 7). On the other hand, we did not see any effect of the composition on this first fracture due to the early rupture of the coating layer at the relative humidity used for the tests (50% RH).

The deposition of the coating on the PLA films reinforced and increased the deformation (elongation at break) of the films (Figure 7). There was an average of 47% elongation percentage for PLA films versus 95%–150% for the coated films. This increase can be explained by a slight plasticization of PLA on the coating side due to water and glycerol migrations into PLA during the coating process [11]. Plasticization of polymers is a mechanism caused by the "insertion" of small molecules (plasticizers) between the polymer chains, which results in lowering its glass transition temperature and increasing its deformation. The polymer then becomes less hard and less brittle, the polymer chains become more mobile and the diffusion is facilitated for other compounds.

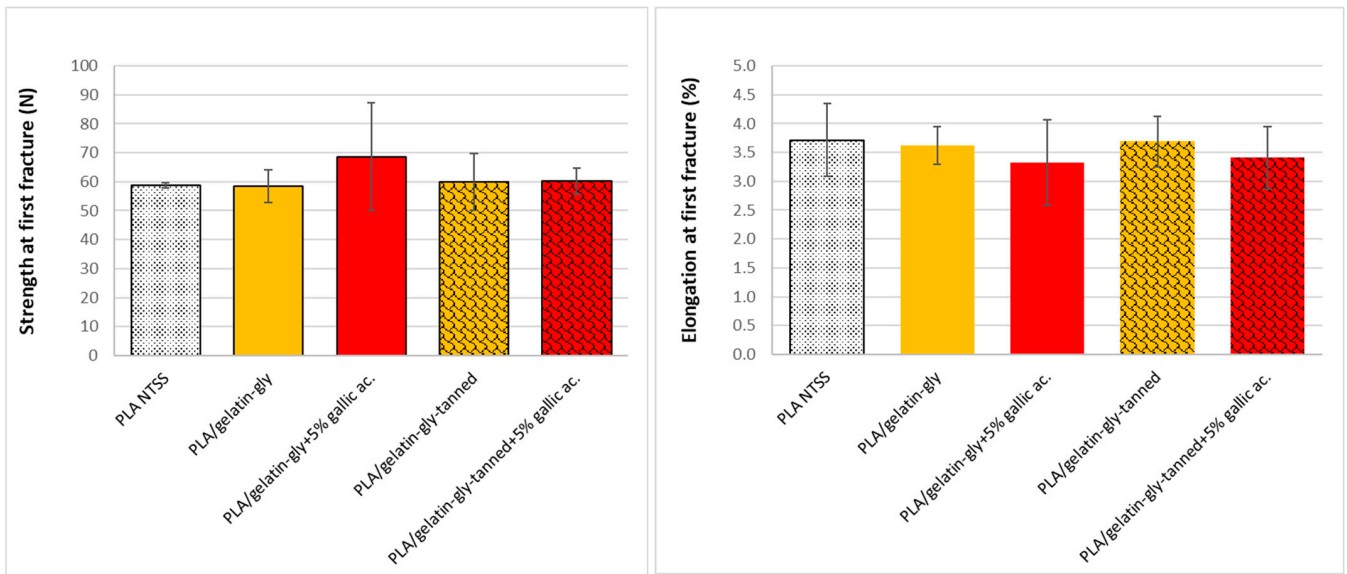

**Figure 6.** Influence of the coating composition on the first fracture (delamination), at 25 °C and 50% RH.

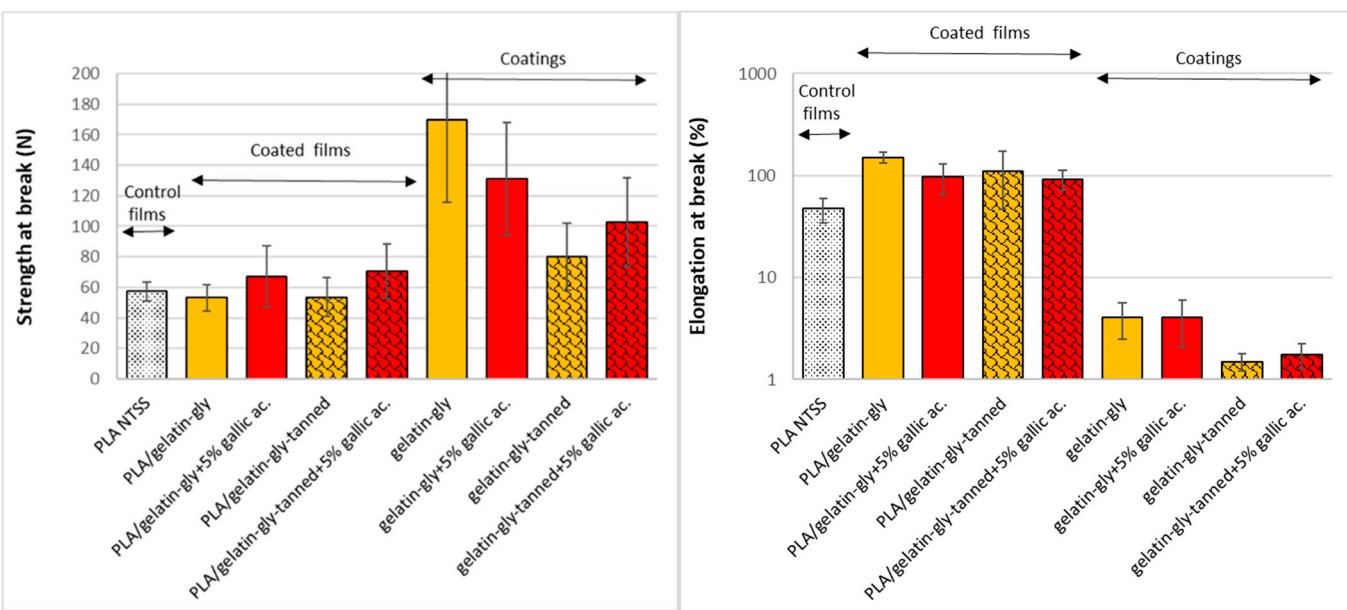

**Figure 7.** Influence of the coating composition on the strength and elongation at breaking, measured at 25 °C and 50% RH.

For the standalone coatings, the percentage was very low in all cases, but it was reduced by the tanning process, from 4% to 1% elongation, and corresponds to the value of the first fracture observed on coated films. On the other hand, the coatings did not significantly modify the strength at breaking, because the thickness of the gelatin was low. The higher values for the standalone coatings can thus be explained by the fact that the coatings are much thicker (150 to 200 µm compared to about 60 µm for the coatings).

*3.5. Film Surface Properties*

The standalone coatings and the coated films showed the same behavior in contact with a drop of water (Figure 8). The two phenolic acids, taken separately, tended (not significant) to increase the contact angle of the coating surface, which would suggest that the surface is more hydrophobic in the presence of gallic or tannic acid. Furthermore, in the absence of gallic acid, a slightly higher value was observed for the tanned film (85.6°) than for the untanned film (78°). However, an opposite effect was observed when both acids were added. There is a (significant) synergistic downward effect between gallic and tannic acids, and the surface wets more easily with water because the angle found is lower, about 60°. This also seems to confirm the rheology observations, which may be due to a less organized and structured network, which in this case presents a less dense and more water-sensitive structure. An interesting element to consider is the spreading rate of the water drop on the surface. This represents the ability of a surface to be wetted quickly or not. It can be seen from the histogram opposite that the treatment of the gels with tannic acid resulted in a speeder spreading rate of the water drop, which confirms that the surface of the tanned films seems to be more hydrophilic. A similar behavior was observed by Etxabide et al. [47] for crosslinked gelatin-based films containing a phenolic compound, tetrahydrocurcumin.

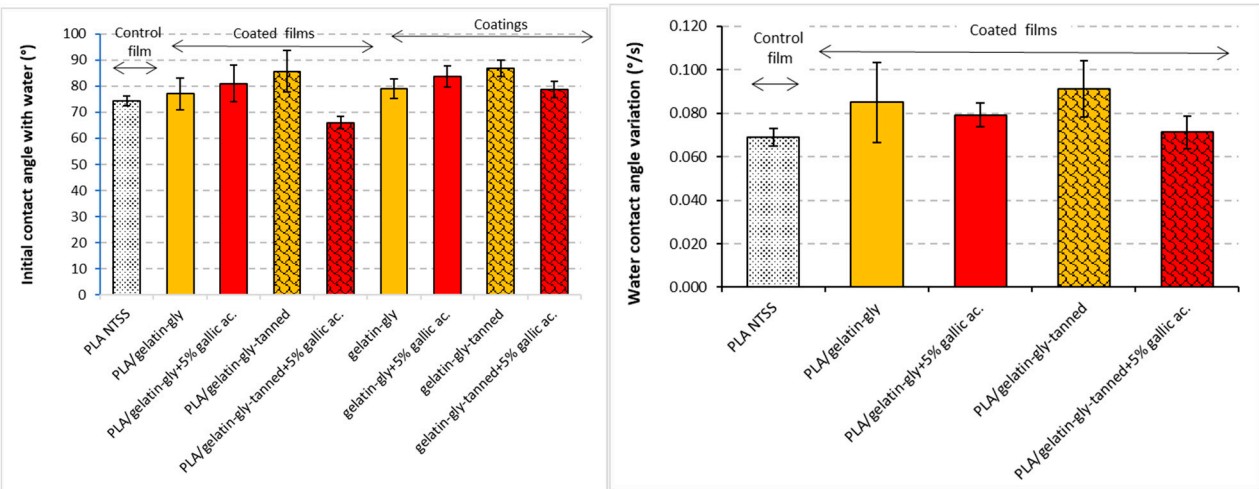

**Figure 8.** Surface properties (initial value and variation of water contact angle) of PLA, standalone coatings and coated PLA films measured at 25 °C and 50% RH.

### 3.6. Film Barrier Properties

When the difference in relative humidity on both sides of the film increases, the rate of water vapor transfer also logically increases. We found a rate of 9.90 $g \cdot m^{-2} \cdot s^{-1}$ for PLA for a difference of 30%–75% RH compared to 14.04 $g \cdot m^{-2} \cdot s^{-1}$ at 30%–100% RH. These results are logical because when the difference in RH increases, the water molecules concentration gradient existing between both sides of the film increases. Furthermore, the higher the humidity level, the more water is absorbed by the films during transfer, which plasticizes the biopolymeric networks of both gelatin and PLA. The positive influence of the relative humidity gradient on biopolymer-based films is well-known and has been demonstrated many times [48,49].

The coating tended to increase the water vapor transfer rate of the PLA film. As shown in Figure 9, at a difference of HR 30–100, the water vapor transfer rate was 16.09 $g \cdot m^{-2} \cdot s^{-1}$ for gelatin-gly-tanned + 5% gallic ac.-coated films versus 14.04 $g \cdot m^{-2} \cdot s^{-1}$ for PLA alone. This can be explained by the fact that the coating induces a plasticization of the PLA, i.e., water enters the film during the coating process, making it more flexible and allowing the polymer chains to be more mobile (molecular diffusion), which facilitates the passage of small molecules. In addition, since the coated side is exposed to the highest humidity, the gelatin layer, which is more water-absorptive than PLA, is protected. Consequently, the coating acts as a reservoir for water adsorbed on the PLA surface, which promotes plasticization and transfer. When the gelatin-based coating is exposed to the lower relative humidity side (30%, PLA/gelatin reverse), it very significantly lowers the rate of water vapor transfer through the film. This experiment confirms the previous conclusion of the reservoir effects due to the gelatin and the plasticization of PLA, which is thus facilitated. In the same trend, Ashwin et al. [50] showed that the coatings of PLA scaffold with a gelatin layer containing or not mucic acid favors PLA wetting and swelling (i.e., plasticization). Pizzoli et al. [43] observed a three-fold increase of the water vapor permeability of PLA/thermoplastic starch-gelatin multilayer films with the amount of gelatin. This behavior was explained by these authors as the consequences the PLA plasticization due to the increase of the water content of the multilayer films with the starch-gelatin layer from 4.51% to 5.44%. The WVTR results only display a weak increase of the moisture transfers through the PLA films that may not significantly affect the water-sensitive food shelf life compare to the neat NTSS-PLA commercial films.

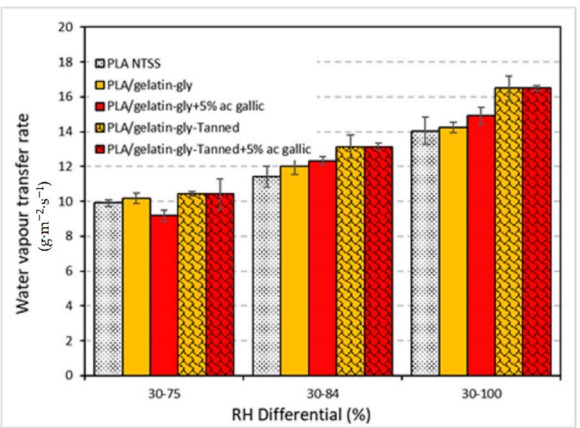 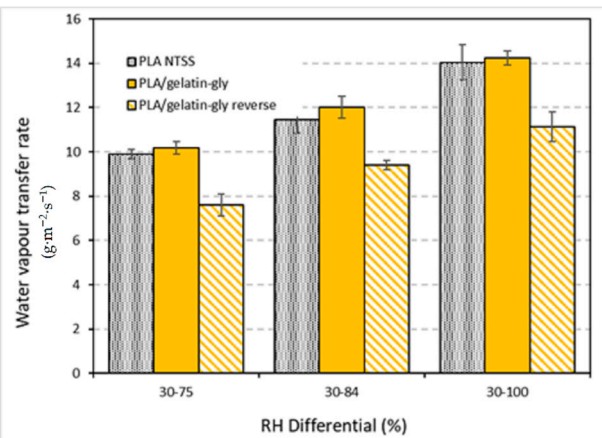

**Figure 9.** Water vapor transfer rate of the PLA and coated PLA films measured for three relative humidity differentials (**left**) and according to the side exposed to the highest humidity (**right**).

Oxygen barrier properties are fundamental for food preservation as much as water barrier properties. However, oxygen is a non-condensable gas and the interaction of films with it can be very different from that with water vapor, which can easily condense and acts as a plasticizer on biomaterials.

The histogram comparing the oxygen transfer rate (OTR) at 85% RH according to the different samples is given in Figure 10. We can see that the coating on PLA films tended to decrease this rate by about 50% (logarithmic scale on the graph). This barrier effect is due not only to the extra layer of gelatin, but also to the fact that at 85% relative humidity, gelatin "soaks up" water, and oxygen is weakly soluble in water. By reducing its solubility, although the PLA and gelatin were plasticized, a decrease in oxygen transfer was observed. Indeed, the solubility of oxygen is lower in water than in polyesters such as PLA. There is thus a competition between water and oxygen in hydrophilic polymers. In addition, one study showed that PLA recrystallized with increasing RH, which also reduces oxygen transfer through the film [11,51]. Standalone coatings are 10 times more effective at slowing oxygen transfer, but this is primarily due to their thickness, which is three to four times greater. However, even though thickness is a factor, correcting for thickness (permeability) also resulted in a better performance of coatings. Gelatin is known to have much better oxygen barrier properties than most biodegradable polyesters [49].

As with water vapor transfer, the effect of the side exposed to the higher humidity have been checked. Thus, Figure 11 shows a very large effect. Oxygen transfer through coated PLA films showed up to 600 times lower oxygen permeation when humidity was reduced (10%), 50 times at 50% RH and only 3 times at 85%. Indeed, at 10% RH, The OTR value was of $9000 \times 10^{-6}$ $cm^3 \cdot m^{-2} \cdot s^{-1}$ for uncoated PLA versus $11 \times 10^{-6}$ $cm^3 \cdot m^{-2} \cdot s^{-1}$ for reverse coated PLA. This reduction in the oxygen transfer rate can be explained by the fact that coating increases the plasticization of PLA and gelatin, which thus favors the passage of $O_2$. In Figure 11, we can also observe a slight decrease in the OTR of PLA with increasing humidity, which may seem contrary since there is plasticization. However, the increase in moisture in PLA also induces recrystallization, and the gas permeability of polymers is very sensitive to the increase in crystallinity [11,52]. Very few studies concern PLA coated with biopolymers and thus it is quite difficult to compare. To relate to the literature, PLA and PLA/gelatin-gly reverse have oxygen permeability of the PLA and coated PLA films at 50% RH of 0.5 and $0.009 \times 10^{-12}$ $cm^3 \cdot m^{-2} \cdot s^{-1} \cdot Pa^{-1}$, respectively. However, considering other PLA coated systems, for instance, coatings based on $SiO_2$ hybrids allowed to only reduce twice the oxygen permeability, decreasing from 1.85 to $0.46 \times 10^{-12}$ $cm^3 \cdot m^{-2} \cdot s^{-1} \cdot Pa^{-1}$ [53]. In another way, Ji et al. [54] demonstrated the detrimental effect of PLA-polyethylene glycol addition into gelatin films on the oxygen permeability that increased from about 0.17 to $2.31 \times 10^{-15}$ $cm^3 \cdot m^{-1} \cdot s^{-1} \cdot Pa^{-1}$, i.e., about 1000 times lower than neat PLA. The reason

could be that the PEG molecule improved the dispersion of the PLA molecule blocks in the gelatin matrix and hindered the formation of PLA phase. Gelatin dominated the properties of the blend film as a continuous phase and PLA affected the properties of the blend film as a dispersed phase.

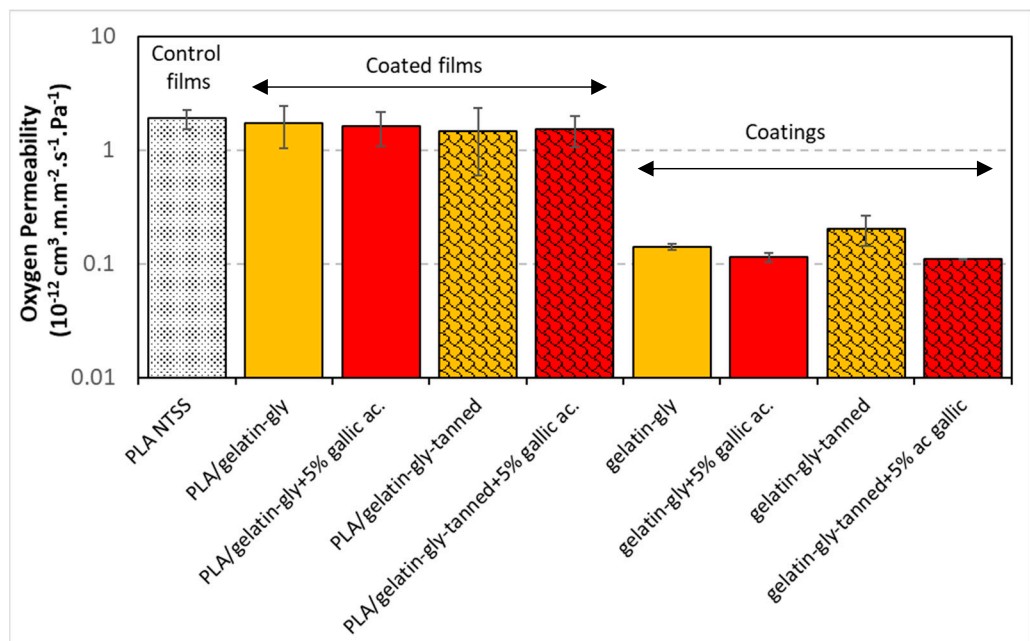

**Figure 10.** Oxygen barrier properties of coated films and coatings measured at 85% RH (coated side exposed to RH = 85%).

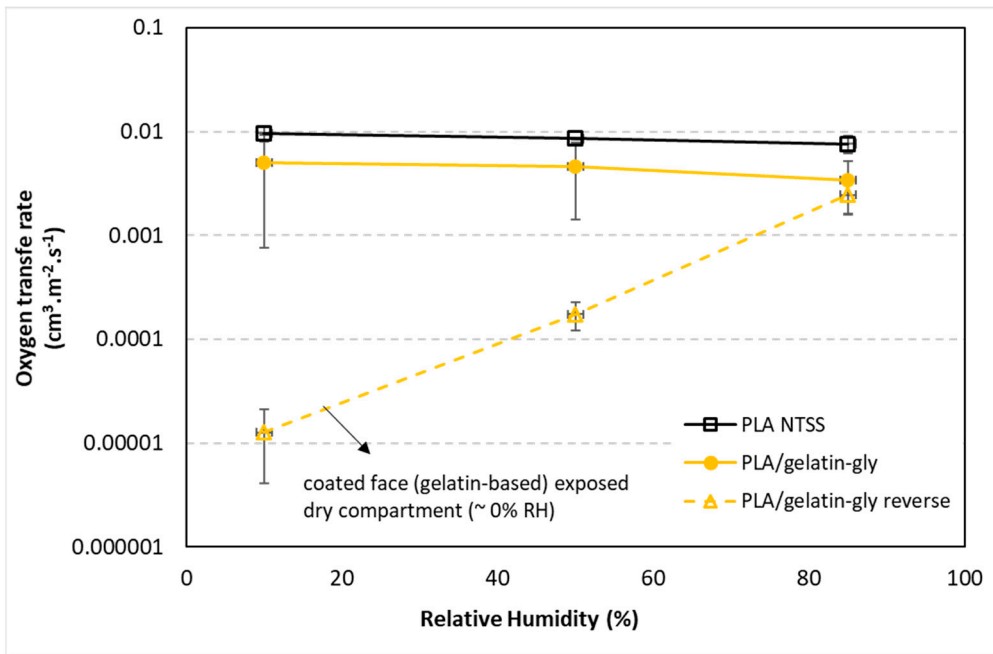

**Figure 11.** Influence of relative humidity and exposed coated side on oxygen transfer of coated and uncoated PLA-based films.

## 4. Conclusions

The aim of this study was to improve mainly the barrier properties of PLA film by applying gelatin suspension coatings in order to allow their use on a wide range of food products by providing antioxidant properties. The results of rheology and mechanical prop-

erties of both the film-forming suspensions and films first gave the intrinsic characteristics influencing the film mechanical properties. Afterwards, following characterization allowed to better understand the film surface (goniometry) and barrier properties (gas permeability, water vapor permeability). The addition of bioactive agents (phenolic acids) in these coatings brought improvements of certain properties of the film. The addition of tannic acid at 0.5%, not for its antioxidant properties, but for its protein tanning (cross-linking) properties did not really lead to an optimization of the properties of the coated PLA films. It is therefore necessary to weigh the pros and cons of these different coatings to know which one would be the most suitable for future use as food films. It may be necessary to consider another way to improve the cross-linking of gelatin, to increase its mechanical strength and reduce its solubility in water or aqueous solutions. However, the gelatin-based coatings, even if they did not contain phenolic acids, allowed to significantly reduce by several order of magnitude the oxygen permeability of PLA films. This will permit to use these films for oils and dry products that are sensitive to oxidation such as biscuits, dry fruits and nuts or cereal-based snacks.

**Supplementary Materials:** The following are available online at https://www.mdpi.com/article/10.3390/coatings12070993/s1, Figure S1: Dynamic rheology curves of the film-forming suspensions during cooling from 50 to 10 °C.

**Author Contributions:** Conceptualization: F.D.; Methodology: F.D., N.B. and C.-H.B.; Investigation: J.R., F.D., N.B. and C.-H.B.; Result analysis and data curation: F.D., N.B. and C.-H.B.; Original draft preparation: J.R. and F.D.; Writing—review and editing: F.D. and C.-H.B.; Supervision: F.D. and N.B.; Funding acquisition: N.B. All authors have read and agreed to the published version of the manuscript.

**Funding:** This work was mainly supported by the own funds of the UMR PAM laboratory, by the Region Bourgogne Franche-Comté (ANER project ActiBioPack—ATP Grant No., 2020-Y-12730) and by the "Fonds Européens de Développement Régional (FEDER)" who invested in lab equipment.

**Institutional Review Board Statement:** Not applicable.

**Informed Consent Statement:** Not applicable.

**Data Availability Statement:** Not applicable.

**Acknowledgments:** The authors wish to sincerely thank Audrey Bentz for English improvement and Adrien Lerbret for his kind availability and help in interpretation of some data related to the rheology. The authors also thank the DIVVA technological platform for the access to rheological, mechanical and surface equipment.

**Conflicts of Interest:** The authors declare no conflict of interest. The data that support the findings of this study are available from the corresponding author upon request.

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
