# Peer review of "Influence of Gelatin-Based Coatings Crosslinked with Phenolic Acids on PLA Film Barrier Properties"

_coatings, doi:10.3390/coatings12070993_

Round 1
Reviewer 1 Report
The presented paper "Influence of gelatin-based coatings crosslinked with phenolic acids on PLA film barrier properties" shows a lot of results and is very interesting and relevant to production plastics (PLA films) biodegradable. The manuscript can be accepted to consider some points:
• line 40 include more references!
• Standardize the images and texts of figure 01
• The formation of films and coatins must be evidenced by a spectroscopic technique (FTIR and/or XPS.).
• Could include rheology curves for the samples studied, perhaps in supplementary material
• Explain the sentence “In the liquid state (40 °C), only tannic acid has an effect on the elastic component (elasticity).”, as it is observed that the variation is within the error of the sample without tannic acid.
• Because figure 05 shows only 3 samples. What is the reason?
• Format the pattern of figures.
Author Response
Reviewer 1 comments :
The presented paper "Influence of gelatin-based coatings crosslinked with phenolic acids on PLA film barrier properties" shows a lot of results and is very interesting and relevant to production plastics (PLA films) biodegradable. The manuscript can be accepted to consider some points:
We thanks the reviewer for the valuable comments and we hereafter expect to give adequate answers and complements to reviewers.
- line 40 include more references!
Some new references added
- Standardize the images and texts of figure 01
Both have been standardized
- The formation of films and coatins must be evidenced by a spectroscopic technique (FTIR and/or XPS.).
The authors do not clearly understand the meaning of this comment. For instance, FTIR allows to demonstrated interactions (shifts) or new bonds (peaks) generated during film formation, but these could also occur and being demonstrated by FTIR in the film-forming suspension prior drying. The fact that films are solid objects able to be manipulated and on which mechanical, surface, barrier properties evidence that a film/coating was formed. On same formulation, FTIR analysis was conducted, as well as the antioxydant properties etc. in a previous paper actually accepted in Journal of Food Science and Agriculture (reference N°17)
- Could include rheology curves for the samples studied, perhaps in supplementary material
We add curves as supplementary data.
- Explain the sentence “In the liquid state (40 °C), only tannic acid has an effect on the elastic component (elasticity).”, as it is observed that the variation is within the error of the sample without tannic acid.
Thanks for noticing the answer does not correspond to data, and we agree that only the two acids-combination has a significant effect on the storage modulus. We revised the sentence in that way.
- Because figure 05 shows only 3 samples. What is the reason?
As stated, all coated films have overlapped isotherm curves. If more curves are added, it is not possible to distinguish in between the 4 coated films. The other isotherms not plotted are availablefor reviewers and readers on demand. However, the GAB parameters displayed in table 1 displayed that there is significant difference among the data of same type of films, coating or coated films (we add mean-difference statistics in the table).
- Format the pattern of figures.
We agree that figure pattern was not homogeneous, and this was due to the shaping by editor into the MDPI publication frame. We reframed some to be more homogeneous.
Reviewer 2 Report
The authors presented interesting studies on PLA films coated with crosslinked suspension of plasticized gelatin incorporating phenolic compounds. I have read this manuscript with great interest, because many of these issues are interesting to me. Essentially, the experiments have been planned and developed in accordance with the rules, and the results are well, explicitly and correctly presented in the paper. Every observed phenomenon is very well described and discussed and the research results are supported by numerous citations. This is well written as well as an interesting and original manuscript. Moreover, this is a very extensive and detailed work. The research results have mainly scientific value.
I think the introduction could be shortened a little. Known facts (like in lines 61-75) can only be mentioned or cited.
In addition, can these materials be easily put into practice? After all, if coated films must be dried for 24 hours, mass production of them can be a problem. Please refer to this in the introduction.
Besides, on page 10, the authors wrote: "In the gel state (at 10 ℃), gallic acid causes a loss of elasticity and stiffness (viscosity) while the tannic acid seems to have no significant effect […] probably due to its low concentration". The word "probably" should not be overused. It was used 5 times in the article. Personally, I think this may indicate some speculation. Maybe it's better to remove that word.
Furthermore, why was 10 times less tannic acid (0.5%) added to the stock solution than gallic acid (5%) according to the same protocol? Please explain it in chapter 2.2. Is the following sentence the only reason for this situation (lines 426-430): „Therefore, TA has 10 carboxyl groups and 25 hydroxyl groups able to interacts with functional groups of gelatins (in adequate pH, O2 concentration and temperature), compared to gallic acid that has only one COOH and three OH groups”?
Moreover, please indicate who is the corresponding author. I think, the authors should also check the literature because there is no formatting consistency.
In my professional opinion no further changes are necessary. I recommend this manuscript for publication after making a few minor revisions.
Author Response
Reviewer 2 comments :
The authors presented interesting studies on PLA films coated with crosslinked suspension of plasticized gelatin incorporating phenolic compounds. I have read this manuscript with great interest, because many of these issues are interesting to me. Essentially, the experiments have been planned and developed in accordance with the rules, and the results are well, explicitly and correctly presented in the paper. Every observed phenomenon is very well described and discussed and the research results are supported by numerous citations. This is well written as well as an interesting and original manuscript. Moreover, this is a very extensive and detailed work. The research results have mainly scientific value.
We really thanks the reviewer for the positive opinion on this work though it has to be improved. We fully agree with the following comments and we expect to give adequate answers or complements in the revised manuscript.
I think the introduction could be shortened a little. Known facts (like in lines 61-75) can only be mentioned or cited.
We tried to shorten the introduction but the continuity and logic of the discussion is lost
In addition, can these materials be easily put into practice? After all, if coated films must be dried for 24 hours, mass production of them can be a problem. Please refer to this in the introduction.
Yes we fully agree that for an industrial application, several hours of drying is not realistic for industrial purpose. We did some trials at pilot scale using flexography where the drying duriation was less than 5 seconds, and thickness about 3-5 microns. Of course permeability reduction was only 10 times reduces, but flexography allows to apply several layers consecutively like for printing different colours. We specified that in thematerials and methods (L208) and in the introduction lines 94 (however, data remains confidential for industrial IP reasons)..
Besides, on page 10, the authors wrote: "In the gel state (at 10 ℃), gallic acid causes a loss of elasticity and stiffness (viscosity) while the tannic acid seems to have no significant effect […] probably due to its low concentration". The word "probably" should not be overused. It was used 5 times in the article. Personally, I think this may indicate some speculation. Maybe it's better to remove that word.
We agree and removed or rephrased.
Furthermore, why was 10 times less tannic acid (0.5%) added to the stock solution than gallic acid (5%) according to the same protocol? Please explain it in chapter 2.2.
We added explanation, that is due to the solubility of tannic acid that is low compared to that of gallic acid. Moreover, the higher efficacy of the tannic acid to crosslink induce a high increase of viscosity that also limit the dispersion and solubility of the tannic acid. . We add some explanation in that way (lines 158, 184 and 190
Is the following sentence the only reason for this situation (lines 426-430): „Therefore, TA has 10 carboxyl groups and 25 hydroxyl groups able to interacts with functional groups of gelatins (in adequate pH, O2 concentration and temperature), compared to gallic acid that has only one COOH and three OH groups”?
Yes we think it is the main reason, more than other physical/chemical characteristics like hydrophobicity etc..
Moreover, please indicate who is the corresponding author. I think, the authors should also check the literature because there is no formatting consistency.
The corresponding author has been specified and is also the main writer of this paper. Literature was checked
In my professional opinion no further changes are necessary. I recommend this manuscript for publication after making a few minor revisions.
Reviewer 3 Report
The article presents a study to mainly improve the barrier properties of PLA film by applying gelatin suspension coatings to allow its use on a wide range of foods, providing antioxidant properties. The results obtained by the authors concluded that the gelatin-based coatings, even if they did not contain phenolic acids, allowed to significantly reduce by several orders of magnitude the oxygen permeability of PLA films. Thus these films could be used for dry products very sensitive to oxidation.
I recommend publishing this article with the observation that
in the introduction to specify more precisely the novelty of this study carried out by them.The article presents a study to mainly improve the barrier properties of PLA film by applying gelatin suspension coatings to allow its use on a wide range of foods, providing antioxidant properties. The results obtained by the authors concluded that the gelatin-based coatings, even if they did not contain phenolic acids, allowed to significantly reduce by several orders of magnitude the oxygen permeability of PLA films. Thus these films could be used for dry products very sensitive to oxidation.
I recommend publishing this article with the observation that
in the introduction to specify more precisely the novelty of this study carried out by them.The article presents a study to mainly improve the barrier properties of PLA film by applying gelatin suspension coatings to allow its use on a wide range of foods, providing antioxidant properties. The results obtained by the authors concluded that the gelatin-based coatings, even if they did not contain phenolic acids, allowed to significantly reduce by several orders of magnitude the oxygen permeability of PLA films. Thus these films could be used for dry products very sensitive to oxidation.
I recommend publishing this article with the observation that
in the introduction to specify more precisely the novelty of this study.
Author Response
Reviewer 3 :
The article presents a study to mainly improve the barrier properties of PLA film by applying gelatin suspension coatings to allow its use on a wide range of foods, providing antioxidant properties. The results obtained by the authors concluded that the gelatin-based coatings, even if they did not contain phenolic acids, allowed to significantly reduce by several orders of magnitude the oxygen permeability of PLA films. Thus these films could be used for dry products very sensitive to oxidation.
I recommend publishing this article with the observation that in the introduction to specify more precisely the novelty of this study carried out by them. The article presents a study to mainly improve the barrier properties of PLA film by applying gelatin suspension coatings to allow its use on a wide range of foods, providing antioxidant properties. The results obtained by the authors concluded that the gelatin-based coatings, even if they did not contain phenolic acids, allowed to significantly reduce by several orders of magnitude the oxygen permeability of PLA films. Thus these films could be used for dry products very sensitive to oxidation.
I recommend publishing this article with the observation that in the introduction to specify more precisely the novelty of this study carried out by them. The article presents a study to mainly improve the barrier properties of PLA film by applying gelatin suspension coatings to allow its use on a wide range of foods, providing antioxidant properties.
The results obtained by the authors concluded that the gelatin-based coatings, even if they did not contain phenolic acids, allowed to significantly reduce by several orders of magnitude the oxygen permeability of PLA films. Thus these films could be used for dry products very sensitive to oxidation.
I recommend publishing this article with the observation that in the introduction to specify more precisely the novelty of this study.
We really thanks the reviewer for the positive opinion on this work though it has to be improved. We fully agree with the following comments and add more clearly the novelty aspects (Line 116)
Round 2
Reviewer 1 Report
The authors have addressed all the concerns raised by the reviewer.